# *Serratia sp.* traits distinguish the lung microbiome of patients with tuberculosis and non-tuberculous mycobacterial lung diseases

Meriem Belheouane[1], Barbara Kalsdorf[2], Stefan Niemann[3,4], Karoline I. Gaede[5,6,7], Christoph Lange[2,4,8,9], Jan Heyckendorf[2¤], Matthias Merker[1,4]*

1 Evolution of the Resistome, Research Center Borstel, Borstel, Germany, 2 Clinical Infectious Diseases, Research Center Borstel, Borstel, Germany, 3 Molecular and Experimental Mycobacteriology, Research Center Borstel, Borstel, Germany, 4 German Center for Infection Research (DZIF), Partner site Hamburg, Lübeck, Borstel, Riems, Borstel, Germany, 5 BioMaterialBank Nord, Research Center Borstel, Leibniz Lung Center, Borstel, Germany, 6 German Centre for Lung Research (DZL), Airway Research Centre North (ARCN), Großhansdorf, Germany, 7 PopGen 2.0 Biobanking Network (P2N), Kiel University, University Hospital Schleswig-Holstein, Campus Kiel, Kiel, Germany, 8 Respiratory Medicine and International Health, University of Lübeck, Lübeck, Germany, 9 Baylor College of Medicine and Texas Children's Hospital, Global TB Program, Houston, Texas, United States of America

¤ Current address: Department of Internal Medicine I, University Medical Center Schleswig-Holstein, Kiel, Germany
* mmerker@fz-borstel.de

## Abstract

### Background

Pathogenic mycobacteria, such as *Mycobacterium tuberculosis* complex (Mtbc), and non-tuberculous mycobacteria (NTMs) can cause severe chronic pulmonary infections. However, not all infected patients develop active disease, and it remains unclear whether key lung microbiome taxa play a role in the pathogenesis of tuberculosis (TB) and NTM lung diseases (LD). Here, we aim to further define the lung microbiome composition in TB, and NTM-LD prior to the initiation of therapy.

### Study design

We employed 16S rRNA amplicon sequencing to characterize the baseline microbiome in bronchoalveolar lavage fluid (BALF) from patients diagnosed with TB (n = 23), NTM-LD (n = 19), or non-infectious inflammatory disease (n = 4). We applied depletion of human cells, removal of extracellular DNA, implementation of a decontamination strategy, and exploratory whole-metagenome sequencing (WMS) of selected specimens.

### Results

Genera *Serratia* and unclassified *Yersiniaceae* dominated the lung microbiome of most patients with a mean relative abundance of >15% and >70%, respectively. However, at the sub-genus level, as determined by amplicon sequence variants

which permits unrestricted use, distribution, and reproduction in any medium, provided the original author and source are credited.

**Data availability statement:** The raw 16S rRNA amplicon-, and the short-read sequences were submitted to the Sequence Read Archive (SRA) under BioProject PRJNA1103672. Reviewer link: https://dataview.ncbi.nlm.nih.gov/object/PRJNA1103672?reviewer=htp8kcmktds7jfcd61p7otgepg R script used to process the 16S rRNA amplicon sequencing data are available at https://github.com/Evobiolo02/Lung-microbiome-of-TB-NTM-patients.

**Funding:** Prof. Niemann, Prof. Heyckendorf, Prof. Lange and Prof. Merker are supported by the German Excellence Cluster for Precision Medicine in Chronic inflammation (EXC2167). Prof. Lange and Prof. Niemann are supported by the German Center of Infection Research under grant agreement TTU-TB 02.709. Prof. Niemann is supported by the Leibniz Science Campus Evolutionary Medicine of the LUNG. PD Dr. Gaede and the BioMaterialBank Nord is supported by the German Center for Lung Research. The BioMaterialBank Nord is member of popgen 2.0 network.

**Competing interests:** PD Dr. Kalsdorf reports personal fees from Insmed Germany GmbH, personal fees from Astra Zeneca , outside the submitted work. PD Dr. Gaede reports grants from German Center for Lung Research (DZL), during the conduct of the study. Prof. Niemann reports grants from German Center for Infection Research, grants from German Excellence Cluster Precision Medicine in Chronic Inflammation EXC 2167, grants from Leibniz Science Campus Evolutionary Medicine of the LUNG (EvoLUNG), during the conduct of the study. Prof. Lange is supported by the German Center of Infection Research. Prof. Lange provided consultation service to INSMED, a company that produced liposomal amikacin as an inhalative suspension for the treatment of NTM-PD OUTSIDE OF THE SCOPE OF THIS WORK. Prof. Lange received speakers honoraria from INSMED OUTSIDE OF THE SCOPE OF THIS WORK. Prof. Lange received speakers honoraria from GILEAD OUTSIDE OF THE SCOPE OF THIS WORK. Prof. Lange received speakers honoraria from Astra Zeneca OUTSIDE OF THE SCOPE OF THIS WORK. Prof. Lange received speakers honoraria from GSK OUTSIDE OF THE SCOPE OF THIS WORK.

(ASVs), TB-patients exhibited increased community diversity, and distinct signatures of ASV_7, ASV_21 abundances which resulted in a significant association with disease state. Exploratory WMS, and ASV similarity analyses suggested the presence of *Serratia liquefaciens*, *Serratia grimesii*, *Serratia myotis* and/or *Serratia quinivorans* in TB and NTM-LD patients.

## Conclusions

The lung microbiome of TB-patients harbored a distinct, and heterogenous structure, with specific occurrences of certain *Serratia* traits. Some of these traits may play a role in understanding the microbial interactions in the lung microbiome of patients infected with Mtbc.

---

## Introduction

Mycobacteria belong to a genus with more than 200 species. Only few species can cause severe respiratory infections. *Mycobacterium tuberculosis* complex (Mtbc) cause tuberculosis (TB), a leading cause of morbidity and mortality worldwide [1]. Although the WHO estimates that approximately a quarter of the global population has been infected with Mtbc, only a minor fraction develops active TB. The main risk factors for developing TB are prolonged contacts to active pulmonary TB patients, immunodeficiencies (*e.g.,* HIV-infection), diabetes mellitus, malnutrition, and tobacco use [1]. Non-tuberculous mycobacteria (NTM), such as *M. avium-complex* or *M. abscessus* can cause NTM lung disease (LD). Transmission of NTMs occurs mostly via environmental sources, direct human-to-human transmission is very rare. Predominant risk factors for NTM-LD are previous pulmonary TB, cystic fibrosis, and bronchiectasis [2,3]. However, the impact of commensal microbes colonizing the lower airways, *i.e.,* the lung microbiome, on pathogenesis of TB and NTM-LD is often neglected.

Few studies have started to address the critical role of the gut and lung microbiome in the onset, progression, and susceptibility to mycobacterial lung diseases [4–7]. Indeed, healthy lungs are not sterile. Landmark studies established the biogeography of the healthy human lung microbiome, and revealed that the upper and lower airways harbor distinct microbiome compositions, while the lower compartment has a reduced microbial biomass [8–10]. Besides, Dickson and colleagues [11] validated bronchoscopy as a reliable method for investigating the lung microbiome and demonstrated that bronchoalveolar lavage fluid (BALF) specimens generate a consistent description of the resident taxa and are less prone to internal and external contamination. Although BALFs generate an adequate picture of the lung microbiome [12], these low-biomass specimens require thorough expertise in microbiome data generation and subsequent analyses [13].

Though previous studies revealed the dysbiosis of the lung microbiome of TB and NTM-LD in distinct patient cohorts; here we aimed to further understand the disparities in lung microbiome structure across TB and NTM-LD, and thus investigated

Prof. Lange is a member of the Data Safety Board of trials from Medicines sans Frontiers OUTSIDE OF THE SCOPE OF THIS WORK. Prof. Heyckendorf and Prof. Merker report a grant from German Excellence Cluster for Precision Medicine in Chronic inflammation (PMI) This does not alter our adherence to PLOS ONE policies on sharing data and materials. There are no patents, products in development or marketed products associated with this research to declare.

BALF specimens collected in a retrospective cohort at the Research Center Borstel, Germany, over 14 years. By directly comparing the lower lung microbiome from BALF specimens, we sought to identify key indicator taxa associated with TB and NTM-LD.

## Materials and methods

### Study design

BALF was routinely collected from patients at the Medical Clinic of the Research Center Borstel (Germany) between July 2003 and September 2017 as part of diagnostic procedures in individuals with radiological abnormalities on thoracic imaging and symptoms of mycobacterial diseases, including fever, night-sweats, weight loss and cough, if mycobacterial DNA had not been detected genotypically in three sputa. The final diagnosis was reached on the basis of BALF microbiology and/or cytology, and radiology results. Bronchoscopy with BALF was performed on these patients according to national guidelines with 200 ml normal saline in fractions of 20 ml each [14]. Sterilizing procedures were performed following national guidelines [15]. For this study, we retrospectively included 63 bio-banked specimens from patients with a confirmed diagnosis of TB, NTM-LD, or non-infectious inflammatory lung disease. Previous medications and secondary diagnoses were retrospectively retrieved from the patient records. The study was positively evaluated by the ethics committee of the University of Lübeck (EK HL AZ 22–249). Patients data were anonymized, processed, and analyzed anonymously.

### 16S rRNA sequencing, and processing

Detailed protocols for depletion of host cells and extracellular DNA, as well as 16S rRNA sequencing of the V3-V4 hypervariable region are provided in the supplementary material. DNA libraries were prepared with a one-step PCR approach using 30 cycles, and sequenced on a MiSeq v3 kit with 2x300bp paired-end reads. We included negative extraction controls, a dilution series of a microbial cell standard, pure cultures from different species, and technical replicates. The resulting fastq files were first processed with dada2 R package (v.1.16.0) [16], then subjected to a thorough multi-step decontamination scheme, and adjustment of clustering thresholds (supplementary material). Finally, sequencing depth was normalized to 10,000 reads for each sample, adjusted ASVs were further clustered into operational taxonomic units (OTUs): 98%, and 97% OTUs (supplementary material, and supplementary S1 Table).

### Ecological and statistical analyses

Statistical analyses were carried out in R (v. 4.2.1) [17]. The mean relative abundances of the main phyla, genera, adjusted ASVs, and 98% OTUs were compared across patient groups using the non-parametric Kruskal–Wallis test. To identify indicator taxa of disease states, we applied indicator value analysis, calculated confidence intervals of the indicator value components, and evaluated the coverage of the identified indicator taxa among disease states in indicspecies" R package (v.1.7.13) [18] (supplementary material).

To explore the interactions between different taxa within patient groups, we calculated Spearman's correlation between the relative abundances of ASVs and 98% OTUs. To evaluate the diversity and community structure of the lung microbiome within, and across patients, we calculated several diversity indices with "vegan" (v.2.6-4) [19], at the adjusted ASV and 98% levels. The within-individual diversity was evaluated by Shannon (observed diversity) and Chao1 (expected richness) indices, and compared across patient groups using the non-parametric Wilcoxon test. To examine community structure, and assess the effect of patient and sampling characteristics along with disease state, we first calculated the Bray–Curtis and Jaccard indices, then applied the non-parametric ANOVA "adonis" with $10^5$ permutations, and constrained principal coordinates analysis. The tested variables were disease group, age group, sex, smoking status, steroid medication, and sampling year.

### Whole-metagenome sequencing (WGS)

We performed an exploratory metagenomic analysis to fine scale the taxonomy of *Serratia* traits. We used BALFs DNA with a concentration >1 ng/μL (NTM patients, n = 3) (detailed methodology in supplementary material).

## Results

### Study cohort

We analyzed 46 out of 63 available BALF specimens which passed the required data quality, and sequencing depth (see supplementary methods). These included adult patients (18 years and older) with pulmonary TB (n = 23), NTM-LD (n = 19), and non-infectious inflammatory lung disease (n = 4) (one sample per patient). To the best of our knowledge, specimens were taken prior to the therapy against the main diagnosis, and as part of the routine diagnosis during an inpatient stay between July 2003 and September 2017. Patient age categories spanned between 10–19 years to 80–89 years, with 60–69 and 70–79 comprising the majority of the patients, *i.e.*, 13 and 14 patients, respectively. Thirty-five percent (n = 16) of the patients were female. Individual TB-patients harbored viral infections, namely, hepatitis and HIV, while some NTM patients were diagnosed with chronic obstructive pulmonary disease (COPD), bronchitis, or pneumonia. Detailed patients' characteristics are provided in supplementary S2 Table.

### Composition of the lung microbiome in patients with pulmonary TB and NTM-LD

To assess the similarities, and disparities of the lung microbiome across patients with pulmonary TB, with NTM-LD, and patients with other pulmonary diseases, we compared the most abundant taxa, and found that *Proteobacteria* strongly and similarly dominated the lungs of all three patient groups (Fig 1A, S1A Fig, and S2 Fig). In addition, we observed no significant differences in the abundances of the major phyla among TB and NTM-LD patients (Wilcoxon rank sum test, p > 0.05). At the genus level, the genera "unclassified *Yersiniaceae*" and *Serratia* (which also belongs to the family *Yersiniaceae*) dominated the lungs of all three patient groups (mean relative abundances of >70%, and >15%, respectively, while the genus *Mycobacterium* was detected in only one NTM patient (Fig 1B, and S1B Fig). Few individuals were highly dominated by a single taxon, such as *Acinetobacter* and *Proteus,* and did not harbor any *Serratia* nor unclassified *Yersiniaceae* (S2A Fig). Overall, the abundances of these main genera were similar across TB, and NTM-LD patients (Wilcoxon rank sum test, p > 0.05).

### Diversity of *Yersiniaceae,* and *Serratia* taxa

To further evaluate the diversity of the lung microbiota, and given the strong dominance of the genera **"**unclassified *Yersiniaceae"* and *Serratia* across all three patient groups, we defined taxa belonging to these genera at the sub-genus resolution level, namely the ASVs, and 98% OTUs. We compared the mean abundances of the major ASVs and 98% OTUs across TB, NTM, and patients with further pathologies. We observed that ASV_1 "unclassified *Yersiniaceae"* and

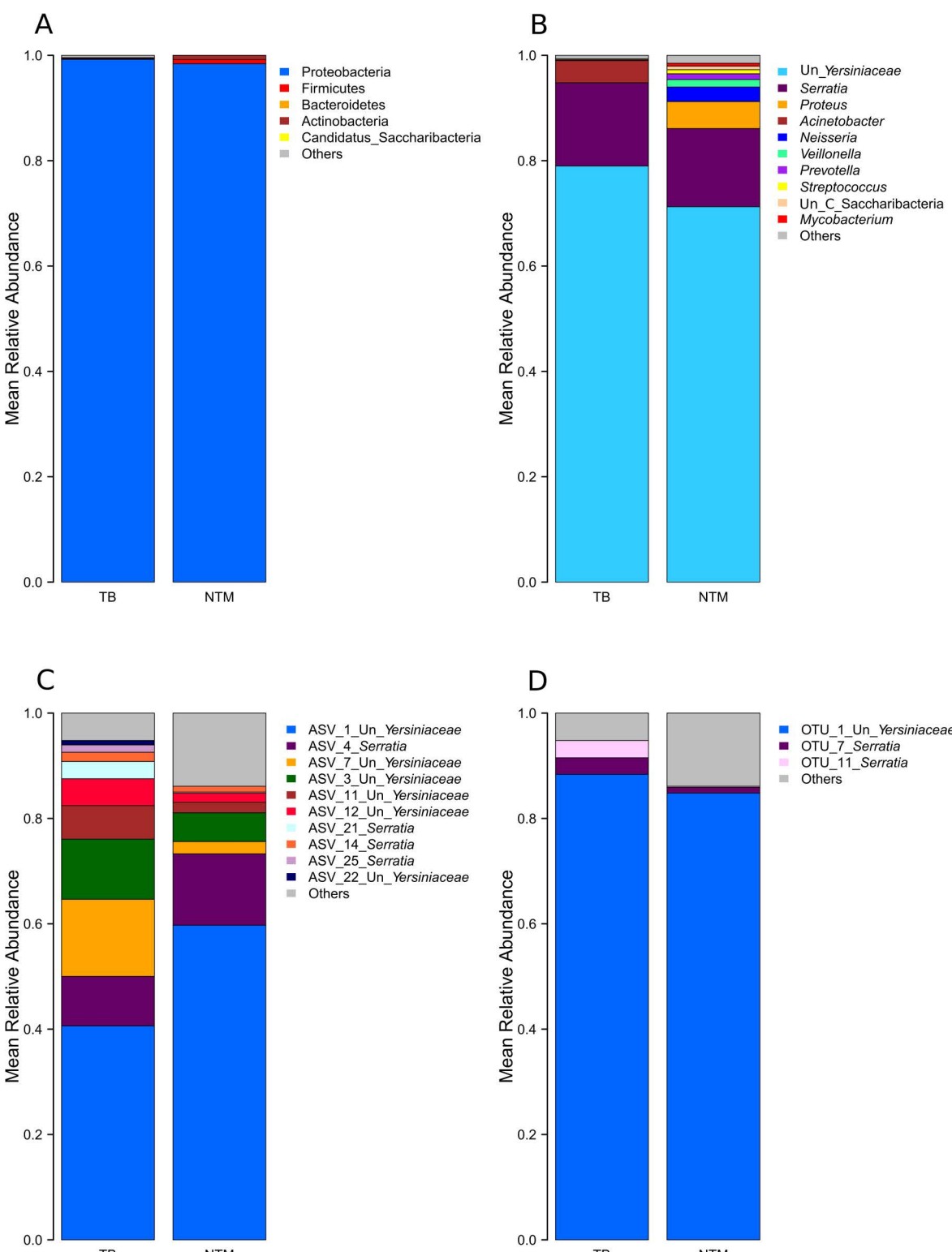

**Fig 1. Mean relative abundances of major taxa across TB and NTM-LD patients. A.** Phyla, **B.** Genera, **C.** ASVs, **D.** 98% OTUs. Un: unclassified.

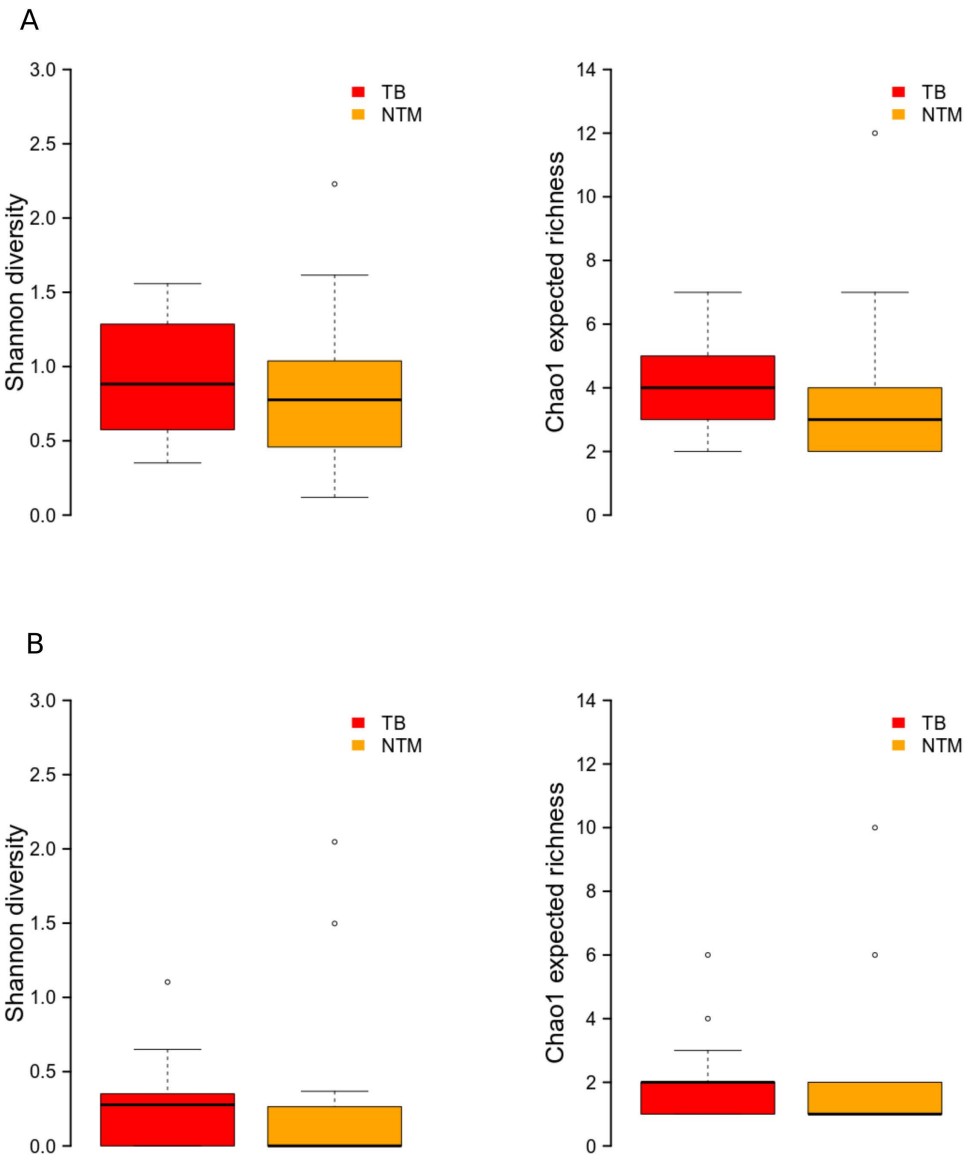

**Fig 2. Shannon and Chao1 diversity indices in TB, and NTM patients.** A. ASVs, B. 98% OTUs.

ASV_4 *Serratia* dominated the communities of all three patient groups (Fig 1C, and S3 Fig). Moreover, TB patients exhibited substantially greater diversity, and significantly higher abundances of ASV_7 unclassified *Yersiniaceae"*, and ASV_21 *Serratia* (Wilcoxon rank sum test, corrected p values: 0.04, and 0.05 for ASV_7, and ASV_21, respectively).

In addition, at the higher clustering level of 98% OTUs, we observed considerably less diversity among patients whereby a single taxon, namely, 0.02_OTU_1 "unclassified *Yersiniaceae"*, dominated all three patient groups (Fig 1D, and S1D Fig), while 0.02_OTU_11 *Serratia* abundances significantly differed across patients (Wilcoxon rank sum test, corrected p value = 0.03). These results suggest that the disparities in unclassified *Yersiniaceae* and *Serratia* traits across patients lay primarily at the species/sub-species level.

Due to a lower sample size (n = 4) of patients with other pulmonary diseases, we explored further aspects of the micro-biome diversity, and structure only in TB and NTM-LD patients. We assessed the within-individual diversity (*i.e.,* alpha diversity), and found similar observed ASVs diversity between TB, and NTM-LD patients, and marginally significant greater expected richness of ASVs in TB-patients (Fig 2A) (Wilcoxon rank sum test, corrected p values: p = 0,14; p = 0.07, for Shannon, and Chao1, respectively). In contrast, 98% OTUs-based diversity indices showed similar observed, and expected richness across patient groups (Fig 2B) (Wilcoxon rank sum test, corrected p values = 0.2814 for Shannon, and Chao1 indices). This result is consistent with our previous observations that TB patients harbored a greater diversity of unclassified *Yersiniaceae* and *Serratia* ASVs.

## Indicator taxa of disease state

To examine potential associations between lung microbiome composition and disease state, we applied "indicator species analyses" to ASVs and 98% OTUs to identify taxa which abundances differed across TB, and NTM patients. First, Indicator value analyses of ASVs revealed that ASV_7 unclassified *Yersiniaceae* and ASV_21 *Serratia* tended to associate with TB patients (corrected p = 0.17 for both ASVs). The same pattern was observed for 0.02_OTU_11 (corrected p = 0.25) (supplementary S3 Table). Specifically, the indicator value components of specificity (A) and fidelity (B) showed that ASV_7 was highly restricted to TB patients (A = 0.86 [95% CI 0.609–1.00]) and occurred in 57% of patients (B = 0.57 [95% CI: 0.360–0.767]). Similarly, ASV_21 was almost exclusively detected in BALFs of TB patients (A = 0.94 [95% CI 0.798–1.00]) and was present in approximately 40% of the individuals (B = 0.39 [95% CI 0.192–0.60]). Furthermore, 0.02_OTU_11 *Serratia* was strongly specific to TB patients (A = 0.94 [95% CI 0.8–0.1] with moderate fidelity B = 0.39 [95% CI 0.192–0.60]).

Second, preference analysis confirmed the indicator value analyses, and revealed that ASV_7, ASV_21, and 0.02_OTU_11 significantly preferred TB-patients (p values: 0.0151, 0.039, and 0.014 for ASV_7, ASV_21, and 0.02_OTU_11, respectively, after Sidak's correction for multiple testing). Overall, 75% the TB-patients carried one or more of the identified ASVs, and 35% harbored the indicator 0.02_OTU_1. Interestingly, these proportions changed across the specificity "A" parameter. For instance, ASV indicators which cover 40% of TB patients, harbored an extremely high specificity to TB patients (A > 0.90) (Fig 3A, 3B, and supplementary S3 Table).

In summary, taxon analyses revealed distinct *Serratia* traits are exclusively identified among specimens from TB-patients. This interesting pattern likely reflect intrinsic disease and/or patient characteristics.

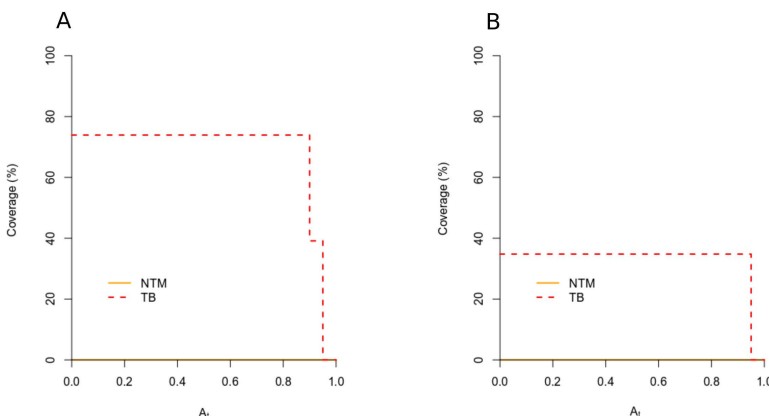

**Fig 3. Coverage of the identified indicator taxa along the "A" threshold. A.** ASVs, B. 98% OTUs.

## Lung microbiome diversity across disease states

To more deeply examine the microbiome structure across disease states, we estimated Bray–Curtis and Jaccard indices (*i.e.,* inter-individual diversity), and assessed the influence of patient, and sampling characteristics. After evaluation of the co-variates (distribution, sample size variations), only disease status, and age category were included in the final models.

In Bray–Curtis model, disease status significantly influenced community structure, while age had a marginally significant influence (*adonis*, disease status: $R^2 = 0.057$, $p = 0.03$; age category $R^2 = 0.219$, $p = 0.089$, $10^5$ permutations). For the Jaccard index, disease status significantly explained a substantially greater portion of the variance compared to the Bray–Curtis index (*adonis*, disease status $R^2 = 0.072$, $p = 0.005$, $10^5$ permutations), suggesting that disparities between TB-and

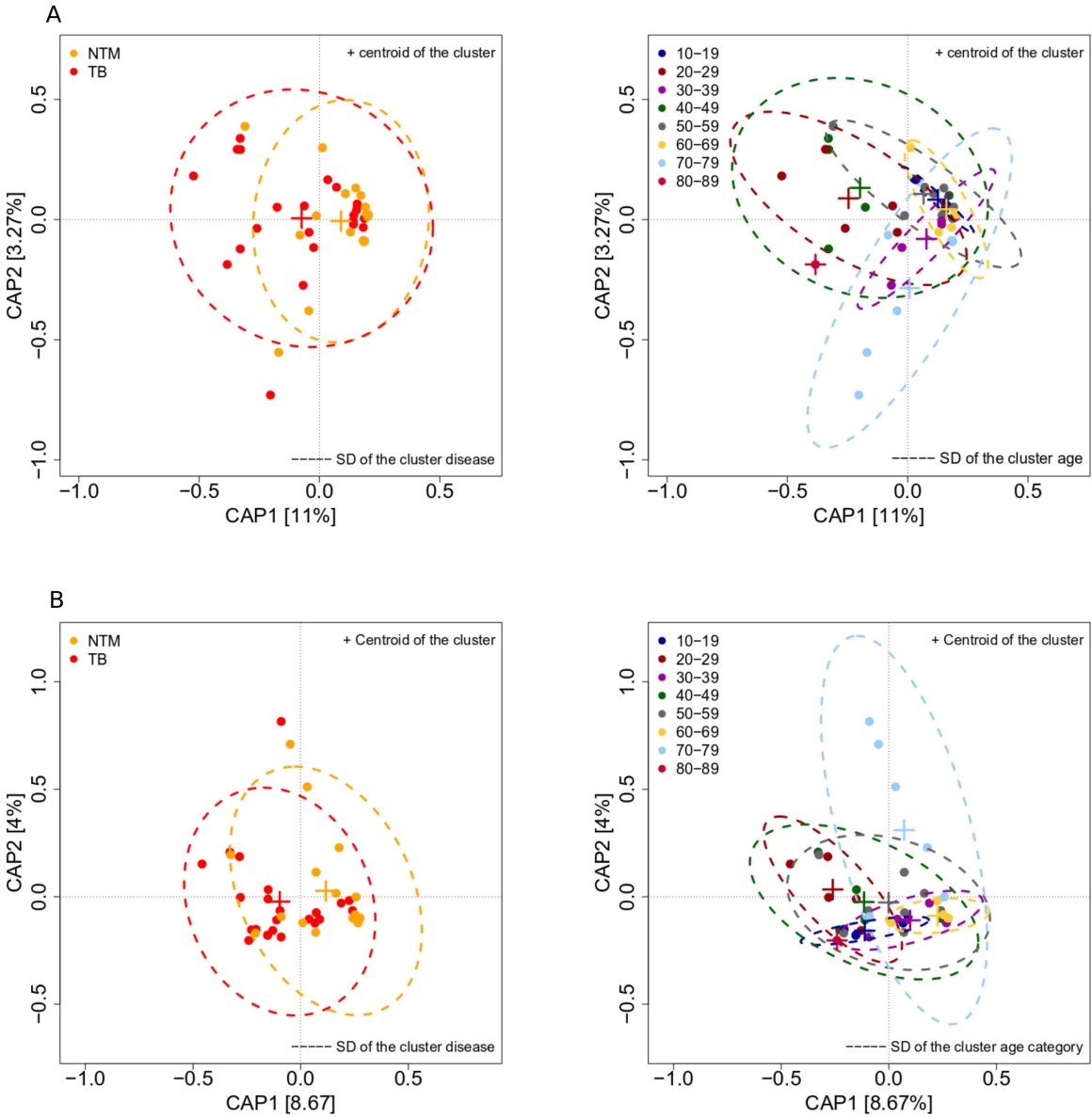

**Fig 4. Constrained principal coordinates analysis of Bray-Curtis (A), and Jaccard (B) indices bases on ASVs.** SD, standard deviation.

NTM patients were driven primarily by the presence/absence of certain ASVs. In contrast, age category did not significantly influence Jaccard index (adonis, age category $R^2 = 0.166$, p = 0.40, $10^5$ permutations).

Furthermore, constrained principal coordinates analyses revealed that i) disease status explained a more substantial proportion of variance in the presence-absence datasets than in the abundance datasets and ii) patients clustered first by disease status, while the influence of further factors, including age category, remained marginal: Capscale, constrained inertia, Bray-Curtis = 0.2403, Jaccard = 0.2215; "anova.cca" by term, disease status: explained variance, Bray-Curtis = 4.6%, p = 0.029, Jaccard = 5.30%, p = 0.004; age category: explained variance: Bray-Curtis = 19.76%, p = 0.09, Jaccard = 17%, p = 0.41, $10^5$ permutations (Fig 4A, and 4B). Of note, analyses on the 98% OTUs indicated influence of disease status remained significant only on the presence/absence datasets whereas community structure was not significantly impacted by disease status, and age category (see supplementary results, and S4 Fig). In essence, most disparities in lung microbiome diversity were evident at the sub-species ASV level, and disease status notably impacted community structure, particularly in presence-absence datasets.

Lastly, we also observed distinct *Serratia* ASV and OTU interaction patterns within each disease state (see supplementary material, S5 and S6 Figs, and supplementary S4 Table). Additional exploratory analyses to determine the species taxonomy suggested that ASV_1, ASV_3, and ASV_7 represent *Serratia liquefaciens* and/or *Serratia grimesii,* and ASV_4, ASV_14, and ASV_21 indicate the presence of *Serratia myotis* and/or *Serratia quinivorans* while the WGS exploratory species assessment mainly indicated the presence of *Serratia grimesii* (see supplementary material, S7 Fig and supplementary S5 Table).

## Discussion

We profiled the lung microbiome of patients with TB, NTM-LD, and non-infectious inflammatory lung diseases, and established a workflow for microbiome analysis of low-biomass pulmonary specimens. We confirmed previous reports showing a high abundance of the genus *Serratia* in BALF specimens. Moreover, we identified distinct *Serratia sp.* traits (*e.g.*, species, subspecies or strains) that characterized, and distinguished the lung microbiome of patients with pulmonary TB, and NTM-LDs. However, it remains to be investigated whether individual strains of *Serratia sp.* and pathogenic mycobacteria interact mechanistically, and whether their co-occurrence in the lower airways influences the onset or progression of the disease.

### *Yersiniaceae* and *Serratia* dominate the lung microbiome of pulmonary patients

We showed that the genera unclassified *Yersiniaceae* and *Serratia* (family *Yersiniaceae*) dominated the lung microbiome of individuals with pulmonary TB, NTM-LD, and non-infectious inflammatory lung diseases. Similarly, previous studies that combined BALF and 16S rRNA amplicon sequencing reported a high prevalence of specific single taxa [20,21]. Specifically, *Serratia* has been described in diverse lung pathologies. Gupta and colleagues [22] reported that *Serratia* dominated the lungs of patients with exacerbated COPD, interstitial lung diseases and sarcoidosis but not in patients with stable COPD, and Fenn et al. 2022 [23] reported that *Serratia* reached 80% relative abundances in ventilator-associated pneumonia (VAP) patients. Moreover, numerous further studies reported an increase of *Serratia* in active disease phases of infectious, and non-infectious disorders in BALF specimens [24–30], and lung biopsies [31]. In a large cohort of individuals with pulmonary TB, Hu et al. [32] identified the genus *Serratia* as the major taxon in the lung microbiome of patients with or without culturable mycobacteria from BALFs. Differences in the microbiome composition before, and after anti-TB therapy, *i.e.*, after culture conversion, were partially driven by two *Serratia* OTUs. The OTUs' abundances decreased after sputum culture conversion, indicating that *Serratia* traits were affected by the presence/absence of Mtbc bacteria and/or the anti-TB medication itself. Notably, Hu et al. [32] and others could not detect the genus *Mycobacterium* in all culture-positive specimens, highlighting the detection limits of 16S rRNA sequencing for these low-biomass specimens [33,34].

**TB-patients harbor a distinguishable heterogeneous lung microbiome structure**

Our data confirmed previous studies, using 16S rRNA amplicon sequencing and BALFs or lung biopsy, that the lung microbiome of TB and NTM-LD patients prior to therapy exhibited greater richness as compared to controls [35–37]. In contrast, Xiao et al. [38] employed WMS and reported increased diversity, and expected richness in cured TB-patients, while patients receiving therapy exhibited intermediate patterns, and active TB-patients showed the lowest diversity, and richness. The start of anti-TB therapy profoundly alters the lung microbiome structure, further decreases potentially beneficial taxa, and selects for antibiotic resistance genes (ARGs) [37,38]. Overall, the observed higher richness in (untreated) TB-patients is likely the consequence of impaired colonization resistance, whereby the healthy homeostatic microbiome prevents the infiltration, and growth of opportunistic pathogens [39–41].

In addition to the elevated taxa richness in TB, our analyses of beta diversity measures confirmed that TB, and NTM-LD significantly influenced community structure prior to therapy, and more strongly on presence/absence, while the impact of patient-related factors, including age, remained marginal. Moreover, we identified specific indicator ASVs for the lung microbiome of TB-patients, and highlighted the higher microbiome heterogeneity of TB- compared to NTM-LD patients. Though previous studies found the dominance, or increase of genus *Serratia* to be common across various lung pathologies, the species, and strains composition of *Serratia*, and their variability across distinct lung diseases is yet to be clarified. Our results based on sub-genus analyses identified disease-specific signatures of the lung microbiome, indicating that while *Serratia* opportunistically colonizes compromised lungs, the species and/or strains diversity, together with interaction patterns within the lung microbiome likely occurs in a disease-specific manner.

The onset, and enrichment of *Serratia* in TB-patients remains to be understood. *Serratia marcescens* is the most investigated species of this genus, and previous studies shed light on how S. *marcescens* interacts with, and evades the immune system. These mechanisms include i) impairment of cellular immunity via production of serralysin metalloprotease [42]; ii) suppression of paralytic peptide (PP) activation via hemocyte killing [43] and iii) the PhoP/PhoQ system [44]. Some of which are shared with the host response to *Mycobacterium tuberculosis* infection. Thus, the enrichment of *Serratia sp.* in TB patients may reflect a scenario in which a weakened immune system—already compromised by *Mycobacterium sp.* infection—allows *Serratia sp.* to act as an opportunistic pathogen. In this context, *Serratia sp.* may evade immune defenses and proliferate as a secondary infection.

Another striking result is the absence of *Serratia* in two patients that were dominated by *Acinetobacter*, and *Proteus*. There is still insufficient knowledge about co-infections of *Mycobacterium sp.* and further pathogens in the lungs, nor about the dynamics of these co-infections. *Acinetobacter baumannii* was defined as the second most frequent co-infection in TB after *Pseudomonas aeruginosa* [45]. Besides, we learnt from previous research that *Serratia* dominates the lungs in the active, and exacerbated phases of the pathologies. Knowing that the exacerbated disease phase such as in COPD is characterized by an increased airway inflammation, and mucus production, it is possible that *Serratia* co-infection takes over the lungs later in the co-infection dynamics; once the airways are further damaged by the original infection, and subsequent co-infection(s). Overall, parameters such as duration of the infection/activation of the disease, severity of airway's inflammation, the host immune components, and microbial metabolites involved in the co-infections dynamics, will help clarify the co-infections sequence in *Mycobacterium sp.* lung pathologies.

Yet, it is unclear if the microbiome and its metabolites influence the activation of TB. A pioneer study from Zhang and colleagues [37] revealed that active TB patients harbored depleted metabolic functions involved in the biosynthesis of lipid metabolism, and increased metabolism of terpenoids and polyketides. Besides, Zhou et al, [46] examined lung biopsies of TB patients, and found *Porphyromonas* associated with TB lesions suggesting *Porphyromonas* might contribute in lesions formation. Thus, future investigations to fully identify the microbial pathways, expressed genes, and functions involved on the onset of TB are warranted.

### *Yersiniaceae* and *Serratia* harbor distinctive traits, with different interaction patterns

Our data further suggested that more than one strain, subspecies or species of *Serratia sp.* was present in the patient lungs microbiome. To date, the mechanisms underlying intraspecies and interspecies interactions within the genus *Serratia* remain poorly understood, and are largely focused on the opportunistic pathogen *Serratia marcescens* [47–49]. Strikingly, pioneer studies characterized the molecular basis of interactions between *S. marcescens* and *M. tuberculosis* and demonstrated a growth-promoting effect of specific siderophores secreted by *Serratia marcescens* in iron-depleted environments. Further cytotoxic assays revealed that these siderophores exhibit cytotoxic activity against human cells [50,51]. Here, we observed that within patients with pulmonary TB and NTM-LD, *Yersiniaceae*/*Serratia* ASVs exhibited differential interaction patterns, with correlations shared across TB and NTM-LD patients, whereas others were unique to a specific disease state. This intriguing pattern likely indicates a dynamic disease-specific interaction driven by growth competition among distinct sub-species and/or strains of *Yersiniaceae*/*Serratia*, as demonstrated for other barrier organs, including the gut and skin [52–54].

## Strengths and limitations

We reveal the disparities, and heterogeneity of the lung microbiome of TB- compared to NTM-LD patients, and present *Serratia sp.* traits as a potential major player in understanding the microbiome dynamics in Mtbc, and NTM-LD.

Due to the retrospective study design, we could not collect samples from negative controls such as saline, oral and bronchoscope washes. Thus, we constructed a rigorous and strict decontamination scheme. Specifically, we assessed the bacterial biomass in the BALF supernatants to detect possible contaminants, *i.e.*, taxa that negatively correlated with the sample biomass and were not detected across technical replicates. In addition, we included several positive controls (mixed, and pure cultures), and technical replicates of positive controls and native BALFs, to optimize the detection of contaminants and spurious taxa, as recommended earlier [13].

This rigorous quality check procedure resulted in a lower sample size, essentially among the control group. Besides, the invasive bronchoscopy sampling method strongly limits the availability of healthy controls 'specimens as previously reported in several studies which investigated BALF microbiome, and thus authors included patients of a non-infectious pulmonary pathology as controls to TB, and NTM diseases. Lastly, the prior medical history of some TB, and NTM patients might be incomplete/unknown, and heterogenous among the patients. This, together with further parameters including the exact infectious strain(s), duration of the infection, physiopathology of the lungs, secondary infections, damaged gut-lung barriers could explain the heterogeneity in the structure, and diversity of *Serratia* traits we reported within TB patients, and across the three different patient groups.

## Conclusions

A better understanding of how some *Serratia* traits impact the dynamics of the lung microbiome in pulmonary TB and NTM-LD could reveal new insights into the course of both diseases. Further work is needed to explore strain-level effects among the genus *Serratia*, microbe–microbe interactions, and resulting implications for disease progression and therapeutic outcomes.

## Supporting information

**S1 Fig. Mean relative abundances of major taxa across the three patient groups.** A. Phyla, B. Genera, C. ASVs, D. 98% OTUs. Un: unclassified.
(TIF)

**S2 Fig. Relative abundances of major phyla, and genera across patient groups.** A. Phyla, B. Genera.
(TIF)

**S3 Fig. Relative abundances of ASVs across patient groups.**
(TIF)

**S4 Fig. Constrained principal coordinates analysis of Bray-Curtis (A), and Jaccard (B) indices baes on 98% OTUs.**
(TIF)

**S5 Fig. Pairwise Spearman's correlation matrix between major ASVs.** A. TB, B. NTM. Only significant correlations after p values correction are shown. Un: Unclassified.
(TIF)

**S6 Fig. Pairwise Spearman's correlation matrix between major 98% OTUs.** A. TB, B. NTM. Only significant correlations after p values correction are shown. Un: Unclassified.
(TIF)

**S7 Fig. Relative abundances of microbial species identified in 3 NTM patients.** Only species with a relative abundance >0.1% are shown, with *Serratia grimesii* occurring in all 3 BALF specimens (from left to right, 7%, 78%, and 16%, respectively).
(TIF)

**S1 Table. Detected, shared, and unique ASVs, adjusted ASVs, and 98% OTUs along the processing steps in the positive controls, and technical replicates.** 1.1. Number of detected ASVs, adjusted ASVs, and 98% OTUs along the processing steps in the pure culture, and mock standard samples. 1.2. Number of detected ASVs, adjusted ASVs, and 98% OTUs along the processing steps in the technical replicates (BALFs, pure cultures, and Mock standard).
(XLSX)

**S2 Table. Description of the patients analyzed in this study TB (n=23), NTM (n=19), Other (n=4).**
(XLSX)

**S3 Table. Indicators taxa analyses based on indicator value analysis (IndVal) on adjusted ASVs (3.1), and 98% OTUs (3.2).**
(XLSX)

**S4 Table. Correlation coefficients, and adjusted p values of pairwise Spearman's correlation between ASVs (1), and 98% OTUs (2) relative abundances in TB, and NTM patients.**
(XLSX)

**S5 Table. Pairwise distances between the main *Yersiniaceae*, and *Serratia* adjusted ASVs, and further *Yersinia*, and *Serratia* previously published strains calculated after sequences alignment in Geneious.**
(XLSX)

## Acknowledgments

We thank Anja Lüdemann, Larissa Mohr, Vanessa Mohr, Maja Mundzeck, Tanja Niemann, Tanja Struve, Silvia Maass, and Franziska Daduna for excellent technical assistance.

## Author contributions

**Conceptualization:** Meriem Belheouane, Jan Heyckendorf, Matthias Merker.

**Data curation:** Meriem Belheouane, Barbara Kalsdorf, Karoline I. Gaede, Christoph Lange, Jan Heyckendorf, Matthias Merker.

**Formal analysis:** Meriem Belheouane, Matthias Merker.

**Funding acquisition:** Stefan Niemann, Karoline I. Gaede, Christoph Lange.

**Investigation:** Meriem Belheouane, Barbara Kalsdorf, Stefan Niemann, Karoline I. Gaede, Christoph Lange, Jan Heyckendorf, Matthias Merker.

**Methodology:** Meriem Belheouane, Barbara Kalsdorf, Stefan Niemann, Karoline I. Gaede, Christoph Lange, Jan Heyckendorf.

**Resources:** Barbara Kalsdorf, Stefan Niemann, Karoline I. Gaede, Christoph Lange, Jan Heyckendorf.

**Supervision:** Matthias Merker.

**Validation:** Matthias Merker.

**Writing – original draft:** Meriem Belheouane, Stefan Niemann, Matthias Merker.

**Writing – review & editing:** Meriem Belheouane, Barbara Kalsdorf, Stefan Niemann, Karoline I. Gaede, Christoph Lange, Jan Heyckendorf, Matthias Merker.

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
