## [Decision Letter · Decision Letter 0]

PONE-D-25-06704*Serratia sp.* traits distinguish the lung microbiome of patients with tuberculosis and non-tuberculous mycobacterial lung diseasesPLOS ONE

Dear Dr. Merker,

Thank you for submitting your manuscript to PLOS ONE. After careful consideration, we feel that it has merit but does not fully meet PLOS ONE’s publication criteria as it currently stands. Therefore, we invite you to submit a majorly revised version of the manuscript that addresses the points raised during the review process.

**ACADEMIC EDITOR: ** We recognize that the manuscript addresses an important area of research and has merit. However, there are some outstanding questions and issues exist as mentioned by the reviewers' comments; every one of these critiques and concerns should be addressed thoroughly in the manuscript at the appropriate section, in addition to the point-by-point response to reviewer's comments. 

We look forward to receiving your revised manuscript.

Kind regards,

Selvakumar Subbian, Ph.D.

Academic Editor

PLOS ONE

Journal Requirements:

Dr. Kalsdorf reports personal fees from Insmed Germany GmbH, personal fees from Astra Zeneca ,  outside the submitted work. Dr. Gaede reports grants from Deutsches Zentrum Für Lungenforschung (DZL),  during the conduct of the study.Dr. Niemann reports grants from German Center for Infection Research, grants from Excellenz Cluster Precision Medicine in Chronic Inflammation EXC 2167, grants from Leibniz Science Campus Evolutionary Medicine of the LUNG (EvoLUNG),  during the conduct of the study.

CL is supported by the German Center of Infection Research. CL provided consultation service to INSMED, a company that produced liposomal amikacin as an inhalative suspension for the treatment of NTM-PD OUTSIDE OF THE SCOPE OF THIS WORK. CL received speakers honoraria from INSMED OUTSIDE OF THE SCOPE OF THIS WORK. CL received speakers honoraria from GILEAD OUTSIDE OF THE SCOPE OF THIS WORK. CL received speakers honoraria from Astra Zeneca OUTSIDE OF THE SCOPE OF THIS WORK. CL received speakers honoraria from GSK OUTSIDE OF THE SCOPE OF THIS WORK. CL is a member of the Data Safety Board of trials from Medicines sans Frontiers OUTSIDE OF THE SCOPE OF THIS WORK.

Reviewers' comments:

Reviewer's Responses to Questions

**Comments to the Author**

1. Is the manuscript technically sound, and do the data support the conclusions?

Reviewer #1: Partly

Reviewer #2: Yes

Reviewer #3: Partly

2. Has the statistical analysis been performed appropriately and rigorously? 

Reviewer #1: Yes

Reviewer #2: Yes

Reviewer #3: Yes

3. Have the authors made all data underlying the findings in their manuscript fully available?

Reviewer #1: Yes

Reviewer #2: Yes

Reviewer #3: Yes

4. Is the manuscript presented in an intelligible fashion and written in standard English?

Reviewer #1: Yes

Reviewer #2: Yes

Reviewer #3: Yes

5. Review Comments to the Author

Reviewer #1: This manuscript investigates the lung microbiome composition in patients with tuberculosis (TB) and non-tuberculous mycobacterial lung diseases (NTM-LD) using 16S rRNA amplicon sequencing and exploratory whole-metagenome sequencing (WMS). The study presents an interesting analysis of bacterial taxa associated with disease states, particularly highlighting Serratia and Yersiniaceae as potential key players. While the research contributes to our understanding of the lung microbiome in TB and NTM-LD patients, addressing several key concerns regarding statistical rigor, interpretation of biological findings, and contextualization within prior literature would greatly enhance the clarity and impact of the study.

Reviewer comments:

1. Statistical Support

The study employs appropriate microbiome analysis methods, including diversity indices, indicator species analysis, and PERMANOVA (adonis test). However, certain statistical limitations may affect the reliability of the conclusions.

- Sample size

The study analyzes 46 BALF samples, comprising 23 from TB patients, 19 from NTM-LD patients, and only 4 from individuals with non-infectious inflammatory lung diseases. Expanding the control group, particularly by including a larger cohort or, if feasible, healthy individuals, would help better differentiate disease-specific microbiome variations from general lung pathology. Strengthening the control group would improve the robustness of comparisons between infectious and non-infectious conditions and provide clearer insights into the unique microbial signatures of TB and NTM-LD.

-Consideration of Confounding Variables and possible limitations

Incorporating potential confounding factors would enhance the interpretability of microbiome differences across patient groups. Key factors that could influence microbial community composition include:

Antibiotic exposure (both past and present), which may significantly alter microbiome diversity; comorbidities (e.g., HIV, hepatitis, COPD, smoking history), which could contribute to microbiome variability; immune status, as host immune responses can shape microbial colonization patterns; while age and sex distributions are acknowledged, further modeling their impact in statistical analyses or discussing their potential influence as a study limitation would add important context to the results and provide a foundation for future TB-microbiome research.

While age and sex distributions are acknowledged, further modeling their impact in statistical analyses could clarify their influence on microbiome diversity and structure. Addressing these aspects, either through statistical adjustments or discussion of their potential effects and therefore limitations of the study, would add important context to the results and pave the way to further analysis in TB-microbiome.

- Interpretation of Microbiome Associations

The observed differences in amplicon sequence variant (ASV) abundance between TB and NTM-LD patients offer valuable insights into microbiome variability. However, additional analyses would help determine whether these differences are functionally relevant, rather than reflective of natural variation or potential contamination.

Since ASV variability alone does not establish a causal or mechanistic role in disease, integrating complementary approaches—such as functional profiling, metagenomic pathway analysis, or validation in independent cohorts—could provide a more comprehensive understanding of the potential role of Serratia and Yersiniaceae in TB and NTM-LD. Additionally, discussing alternative explanations for microbial shifts (e.g., environmental influences) would help ensure that conclusions are well-supported.

2. Interpretation of Biological Findings

The study provides an intriguing analysis of Serratia and Yersiniaceae differences between TB and NTM-LD patients. Strengthening the discussion around the biological significance of these findings would further enhance the impact of the study. Clarifying whether these differences reflect disease-specific microbiome shifts, opportunistic colonization, or external factors would help frame the results in a broader microbiological and clinical context.

2.1 Potential Contamination in Low-Biomass Samples

Since Serratia and Yersiniaceae are known to be prevalent in laboratory and hospital environments, it would be helpful for the manuscript to briefly discuss measures taken to rule out contamination and further contextualize the robustness of the findings. While the study employs a rigorous decontamination strategy, low-biomass samples like BALF are particularly susceptible to external microbial influences, which could have an impact on microbial abundance estimations. (doi: 10.1186/s40168-016-0172-3; 10.1186/s41479-018-0051-8).

Providing additional details on how negative controls were used to differentiate true biological signals from possible contaminants—or discussing any limitations in this regard—would reinforce confidence in the observed differences. Given that Mycobacterium was detected at low frequency despite being the primary pathogen, addressing whether the observed Serratia/Yersiniaceae abundances could have been affected by technical or environmental factors would add further clarity.

2.2 Clarifying the Functional and Pathogenic Role of Serratia in TB

The identification of Serratia species in TB and NTM-LD patients is an interesting finding, and expanding the discussion on its potential biological role could greatly enrich the manuscript. Since different Serratia species exhibit diverse functional properties, ranging from commensalism to opportunistic pathogenicity, further elaboration on possible mechanisms of interaction with Mycobacterium or host immune responses would enhance the interpretation of these findings.

To strengthen the study’s conclusions, the authors might consider:

- Discussing whether Serratia enrichment in TB patients could be linked to immune modulation, microbial competition, or metabolic interactions within the lung microbiome.

- Exploring alternative explanations for the presence of Serratia, such as potential environmental exposure or prior antibiotic use.

- Acknowledging that functional analyses (e.g., metagenomics, transcriptomics) would be needed to establish Serratia’s role in disease progression, and highlighting this as a potential avenue for future research.

3. Discussion of Prior Literature and Knowledge Gaps

The manuscript would benefit from further contextualization within the broader field of TB microbiome research. A clearer discussion of how this study builds upon prior findings or what knowledge gap it addresses would help position it more effectively within the existing literature. It would be better to clarify (as early as the Introduction) whether this is the first study of the TB lung microbiome using BALF or whether previous studies have already explored similar microbial shifts. Several other studies have characterized TB-associated microbiome changes—yet the discussion does not compare the current findings to those results.

Additionally, further exploration of whether Serratia has been previously reported in TB lung microbiomes (or any other lung condition) would strengthen the discussion by clarifying whether its presence is novel or aligns with existing knowledge. If Serratia’s role in the TB microbiome remains uncertain, acknowledging potential explanations—such as environmental factors, host immune interactions, or sample type differences—could provide a valuable perspective.

Finally, placing these findings within the broader landscape of respiratory microbiome research, including comparisons to conditions like pneumonia or chronic obstructive pulmonary disease (COPD), would offer deeper insights into whether the observed microbiome shifts are specific to TB or part of a more general lung disease signature

Expanding the discussion to explore potential mechanisms by which microbiome changes might influence TB disease progression would significantly enhance the impact of this study. Integrating insights into host-microbe interactions, immune responses, or microbial competition within the lung environment could provide a more comprehensive interpretation of the findings.

Additionally, incorporating established concepts such as colonization resistance, dysbiosis, or microbial metabolic interactions would help frame the observed microbiome shifts within well-recognized biological frameworks. This could strengthen the discussion by offering potential explanations for why specific taxa, like Serratia and Yersiniaceae, may differ between patient groups.

4. Technical Clarifications

4.1 Data Availability

The raw sequencing data and processing scripts are provided; however, limited access to clinical metadata may constrain full reproducibility and comprehensive interpretation.

4.2 Terminology

Since 16S rRNA sequencing is primarily used for bacterial taxonomic profiling, it would be helpful to clarify whether the metagenomic analysis refers to true whole-metagenome sequencing (WMS) or an extended 16S-based bacterial "metagenome" approach. Ensuring accurate terminology would improve the precision of the methods section and the interpretation of the results.

Reviewer #2: In this manuscript, the authors have studied retrospective samples of the lung microbiome of TB, NTM and non-infectious inflammatory disease patients using 16S rRNA amplicon sequencing as well as Whole Metagenome Sequencing. They have found that while these samples are primarily dominated by unclassified Yersiniaceae and Serratia with similar abundance across the disease groups, the sub-genus level analysis indicates difference in abundance certain strains across the groups as well as significant association of certain strains (ASV_7, ASV_21) with the disease state. Overall, the manuscript seems sound and requires minor corrections.

1. Given that the samples were collected retroactively, is it possible that the Serratia and unclassified Yersiniaceae strains evolved over time, especially for TB, as the disease progressed? The manuscript does not indicate when the TB was detected in the patient and how long the patient had it before sampling. The evolution and changes to the underlying strains might be interesting to investigate.

2. What is the corrected p value threshold for indicator taxa analysis? The corrected p-values have been provided but a threshold has not been given, that indicates significance of the analysis.

3. The quality of figures in the manuscript is low; however, once downloaded, the resolution is much better.

4. There are minor grammatical errors in the supplemental section of the manuscript:

a) Line 123: "calculated based"

b) Line 133: "which null hypothesis is"

c) Line 164: "if a within a given sample"

d) Line 177: "which DNA concentration"

e) Line 178: grammatical structure of the sentence needs to be corrected

f) Line 183: "randomly 10 000 reads"

g) Line 189: "which pairwise distance"

5. What database was used for taxonomic assignment for whole-metagenome sequencing? This has not been mentioned in the supplemental section

6. Lines 246 and 249 in the supplemental section - has corrected p-value been used for Kruskal Wallis test? Unlike lines 251 and 255, this has not been mentioned/clarified.

Reviewer #3: The manuscript "Serratia sp. traits distinguish the lung microbiome of patients with tuberculosis (TB) and

non-tuberculous mycobacterial (NTM) lung diseases" describes the composition of the lung microbiota of patients with tuberculous mycobacterial infections and non-tuberculous lung disease. The authors aim to define the compositions prio to the initiation of therapy but also to unravel the role of the microbiome as the initiator or progessor of the development of mycobacterial lower respiratory infections.

The study is technically well performed but lacks some in-depth interpretation of the data. The authors use the data of twoo patients groups only: TB vs NTM lung disease. As the authors mention at page 13, the study lacks a group of healthy controls although it is clear to obtain such samples beacause of ethical reasons. However, it would be worthwhile to present the data of the four non-infectious patients. Sure, the statistical value would be very low due to the low number of non-infectious inflammatory disease, but might underline the typical presence of Serratia and (unclassified) Yersiniaceae in the other groups.

Another question is raised by the fact that some individuals do not harbor any Serratia and/or (unclassified) Yersiniaceae but are colonized/infected by Acinetobacter or Proteus. Which underlying mechanism is superior to Serratia or (unclassified) Yersiniaceae that might give the an ecological advantage that is probably present in some of the Acinetobacter of Proteus spp.?

Technically, the numbers of reads are high enoough to get the resolution to detect low abundance bacteria. Whether the V3-V4 region of the 16S rRNA-gene is sufficient is part of discussion, as it might result to the determination at genus- of subgenus level rather than at species level.

It is advisable to expand the vision of the authors to the issues raised above.

6. PLOS authors have the option to publish the peer review history of their article (what does this mean? ). If published, this will include your full peer review and any attached files.

**Do you want your identity to be public for this peer review?** For information about this choice, including consent withdrawal, please see our Privacy Policy .

Reviewer #1: **Yes: ** SUSANA DE LA TORRE-ZAVALA

Reviewer #2: No

Reviewer #3: **Yes: ** Wil A. van der Reijden

---

## [Author Response · Author response to Decision Letter 1]

25 Apr 2025

We provide a repsonse letter as an attachment for the Reviewers

---

## [Decision Letter · Decision Letter 1]

*Serratia sp.* traits distinguish the lung microbiome of patients with tuberculosis and non-tuberculous mycobacterial lung diseases

PONE-D-25-06704R1

Dear Dr. Merker,

We’re pleased to inform you that your manuscript has been judged scientifically suitable for publication and will be formally accepted for publication once it meets all outstanding technical requirements.

Kind regards,

Selvakumar Subbian, Ph.D.

Academic Editor

PLOS ONE

Additional Editor Comments (optional):

Reviewers' comments:

Reviewer's Responses to Questions

**Comments to the Author**

1. If the authors have adequately addressed your comments raised in a previous round of review and you feel that this manuscript is now acceptable for publication, you may indicate that here to bypass the “Comments to the Author” section, enter your conflict of interest statement in the “Confidential to Editor” section, and submit your "Accept" recommendation.

Reviewer #1: All comments have been addressed

Reviewer #2: (No Response)

Reviewer #3: All comments have been addressed

2. Is the manuscript technically sound, and do the data support the conclusions?

Reviewer #1: Yes

Reviewer #2: Yes

Reviewer #3: Yes

3. Has the statistical analysis been performed appropriately and rigorously? 

Reviewer #1: Yes

Reviewer #2: Yes

Reviewer #3: Yes

4. Have the authors made all data underlying the findings in their manuscript fully available?

Reviewer #1: Yes

Reviewer #2: Yes

Reviewer #3: Yes

5. Is the manuscript presented in an intelligible fashion and written in standard English?

Reviewer #1: Yes

Reviewer #2: Yes

Reviewer #3: Yes

6. Review Comments to the Author

Reviewer #1: The authors have addressed the critical points raised during the initial review. They have enriched the discussion with broader contextualization of Serratia’s role in lung pathophysiology and the implications for TB pathogenesis. Authors explicitly acknowledge the limitations imposed by the low DNA yields from BALF specimens, and justify the robustness of the conclusions in light of these limitations.

Notably, the updated discussion now covers potential mechanisms of microbial interaction, includes pertinent literature, and discusses the strain-level heterogeneity of Serratia as a potential diagnostic or ecological indicator. The response to concerns about experimental robustness is well supported by detailed quality control procedures, contamination filtering, and use of control standards.

The statistical analysis is appropriate, transparent, and well-documented. While some p-values were marginal, the authors are cautious not to overinterpret these findings and clearly present them as exploratory or hypothesis-generating.

The manuscript is written clearly and the presentation is generally strong. I found the figure legends to be informative and the supplementary materials comprehensive.

Reviewer #2: (No Response)

Reviewer #3: The authors have given more insight into the microbial composition of the control patients, actually the "Others" as denoted in Figure S1 and S2. Especially Figure S1 B and C is visually helpful to show the typical composition of the NTM-LD patients.

A further disussion of these findings in line 290-302 has been added which is helpful to interprete these differences between TB, NTM-LD and "Others" patient groups.

Using the V3-V4 part of the 16S rRNA-gene is as suggested limited for taxonomic analyses of a mixed flora. The technical opposite: WGS metagenomics is of course hard to interprete because of the need of extreme high numbers of reads in combination of sufficient biomass as is discussed.

7. PLOS authors have the option to publish the peer review history of their article (what does this mean? ). If published, this will include your full peer review and any attached files.

**Do you want your identity to be public for this peer review?** For information about this choice, including consent withdrawal, please see our Privacy Policy .

Reviewer #1: **Yes: ** Susana De la Torre Zavala.

Reviewer #2: No

Reviewer #3: No

---

## [Editor Report · Acceptance letter]

PONE-D-25-06704R1

PLOS ONE

Dear Dr. Merker,

I'm pleased to inform you that your manuscript has been deemed suitable for publication in PLOS ONE. Congratulations! Your manuscript is now being handed over to our production team.

Kind regards,

on behalf of

Dr. Selvakumar Subbian

Academic Editor

PLOS ONE